

# Sex- and age-related morphological and functional differences in the skull of *Eira barbara* (Linnaeus, 1758) (Carnivora, Mustelidae)

Fernando Araujo Perini[1] and Fernando L. Sicuro[2,3,4]

[1] Departamento de Zoologia, Universidade Federal de Minas Gerais, Belo Horizonte, Minas Gerais, Brazil
[2] Instituto de Biologia Roberto Alcantara Gomes, Universidade do Estado do Rio de Janeiro, Rio de Janeiro, Rio de Janeiro, Brazil
[3] Centro de Estudos do Ambiente e Mar, Universidade de Aveiro, Aveiro, Aveiro, Portugal
[4] Programa Internacional para Ciência, Tecnologia e Inovação em Saúde, Creative Science Park, Ílhavo, Aveiro, Portugal

Corresponding author
Fernando Araujo Perini,
faperini@ufmg.br

## ABSTRACT

**Background:** Morphological differences related to age and sex have been extensively documented in Nearctic and Palearctic mustelids, largely due to the interest in the commercial management of species such as martens and fishers. However, sexual dimorphism and ontogenetic development in Neotropical species, such as the tayra (*Eira barbara*), remain poorly studied.

**Methods:** We investigated the skull development and sexual differences in *E. barbara* through qualitative and morphometric analyses of 376 skull specimens from museum collections. Specimens were classified into four age classes based on tooth eruption, cranial suture closure, and skull morphology. Linear measurements were used to assess biomechanical parameters related to the masticatory function, including temporal and masseter muscle force indices.

**Results:** The results show that male and female tayras follow two distinct growth pathways. While juveniles (Age Classes I–II) share nearly identical skull proportions, males rapidly surpass females in size and robustness by the subadult stage (Age Class III). Fully mature males (Age Class IV) develop broader zygomatic arches, thicker sagittal crests, and significantly stronger bite forces driven by enlarged temporal muscles. On the other hand, *E. barbara* adult females exhibit a narrower post-orbital constriction and palatal region which denotes distinct ontogenetic trajectories between males and females. Multivariate statistical analyses (principal component analysis/discriminant function analysis) confirmed these morphological trends, highlighting allometric growth patterns and functional differences in masticatory biomechanics. Males exhibit greater bite force, particularly in the temporal muscle system, which may be linked to territorial defense or mating behaviors. These findings align with patterns observed in other Guloninae species, such as fishers (*Pekania pennanti*) and wolverines (*Gulo gulo*), but highlight unique aspects of *E. barbara*'s tropical ecology, including non-seasonal breeding, flexible growth patterns, and broad habitat use. Additionally, tayras' asynchrony in sexual maturity and sexual dimorphism in skull biomechanics may be related to distinct ecological roles for the sexes. We hypothesize that these differences are adaptive responses to

the dynamic and resource-rich environments of the Neotropics. Our work underscores the role of ontogenetic and sexual variation in understanding the evolutionary and ecological adaptations of Neotropical mustelids, providing new light on *E. barbara*'s morphology and masticatory function.

## INTRODUCTION

Establishing age and sex in wild animals is essential for wildlife management and life history studies, especially in some carnivore species, in which sex differences incur in marked divergent aspects of their biology (*Isaac, 2005*; *Lindenfors, Gittleman & Jones, 2007*). Mustelid species present several degrees of sexual dimorphism (*Macdonald & Newman, 2017*). Some species, such as small hypercarnivorous species as stoats and weasels, have marked differences between sexes, while others, as somewhat social species as otters and badgers, seem to have little to no differences (*Moors, 1980*; *Gittleman & Valkenburgh, 1997*; *Noonan et al., 2016*). As observed in other Carnivora species, sex differences may even vary intraspecifically among mustelids, in which different populations and taxonomic unities within a species show different degrees of sexual dimorphism (*Isaac, 2005*; *Sicuro & Oliveira, 2015*). However, most of the studies on mustelid sexual dimorphism and ontogenetic development address the Holarctic species (*e.g., Mustela nivalis*, *King, 1980*; *Gulo gulo*, *Wiig, 1989*; *Pekania pennanti*, *Bryant, McGillivray & Bartlett, 1997*; *Martes martes, Martes foina*, *Ruette et al., 2015*; *Baumann & Gornetzki, 2017*; *Özen, 2020*), with tropical species being neglected in this aspect.

Despite ethical issues concerning poaching, hunting, and even legal farming activities, many carnivorans, particularly mustelids, are the focus of great interest in the fur market. Therefore, age and sex determination are crucial to hunting/farming management, defining which specimens could be caught and which subproducts harvested, especially in Northern temperate countries, where the fur industry plays a relevant economic role (*Helldin, 2000*; *Fryxell et al., 2001*; *Milligan et al., 2025*). Therefore, age and sex determination are mostly well known in fur-bearing carnivore species, including martens, fishers, and weasels, since this is key information to modeling population dynamics and abundance; on the other hand, this knowledge is broadly applied in management and conservation (*King, 1980*; *Albayrak, Özen & Kitchener, 2008*; *Ruette et al., 2015*). Not only do the fur industry and wild predator management result in more studies, but these areas also attract more researchers and improve wildlife management policies. However, a reliable system of sex and age determination is also crucial for many studies that rely on morphological variation between and within populations and species, especially in studies of ecology and evolution (*Ruette et al., 2015*).

In most of the Neotropical region, there is no tradition of economic exploration of fur from native carnivorans in the clothing industry. This lack of a well-established fur industry reduces the economic demand for systematic wildlife management of mustelids

and other carnivores. Consequently, there is considerably less information about ontogenetic changes in morphology and sexual dimorphism for Neotropical mustelids. For instance, the tayra, *Eira barbara* (Linnaeus, 1758), is a widespread Neotropical mustelid carnivoran from the subfamily Guloninae and, therefore, related to martens, fishers, and wolverines (*Proulx & Aubry, 2017*; *Law, Slater & Mehta, 2018*). It occurs from southern Mexico to northern Argentina, occupying mainly tropical and subtropical areas, including relatively open environments and anthropogenic habitats, but generally associated with forests (*Villafañe-Trujillo & López-González, 2024*). Tayras are usually considered opportunistic generalists, feeding on fruits and small vertebrates (*Presley, 2000*; *Larivière & Jennings, 2009*).

Fur-bearing mustelids, especially Guloninae species such as martens and fishers, have their postnatal development well studied with many proposals for distinguishing age classes and genders (*Marshall, 1951*; *Bryant, McGillivray & Bartlett, 1997*; *Heptner et al., 2001*; *Albayrak, Özen & Kitchener, 2008*; *Özen, 2020*). Despite the close phylogenetic relation with other fur-bearing mustelids and occasional hunting for local trading and cultural purposes (*Villafañe-Trujillo & López-González, 2024*), the tayra has negligible economic importance as a commercially harvesting species. Color variants in mustelids often drive economic interest and persecution of wild species. Tayras, however, seem to be out of the radar of the fur industry despite their highly variable coat color patterns (*Cotts et al., 2024*). Consequently, compared to other Guloninae, many aspects of its life history are still relatively unknown, except for a few isolated descriptions of captive tayras' external physical and behavioral ontogenetic development (*Poglayen-Neuwall & Poglayen-Neuwall, 1976*; *Panizzon & Azevedo-Filho, 2019*). However, there is a lack of data on the differentiation of the skulls of young and between males and females. Some articles briefly indicate that males are larger and more robust than females (*Presley, 2000*; *Schiaffini, 2020*), a mustelid's typical pattern (*Buchalczyk & Ruprecht, 1977*; *Schmidt, 1992*; *Heptner et al., 2001*), but none of these studies are based on sound quantitative and qualitative analyses. On the other hand, despite the existence of several studies addressing the mandibular functionality of mustelids within evolutionary and ecomorphological contexts (*e.g.*, *Radinsky, 1981a*, *1981b*; *Dessem & Druzinsky, 1992*; *Dumont et al., 2016*; *Law, Slater & Mehta, 2018*; *Law & Mehta, 2019*; *Hartstone-Rose, Hertzig & Dickinson, 2019*; *Gálvez-López & Cox, 2022*), none of these have focused sex and age variations on *E. barbara*'s skull-jaw biomechanics. Functional approaches aim to add an extra layer of complexity to morphological studies, allowing for a better understanding of species' evolutionary trends and the interaction between their anatomy and occurrence in specific habitats. The same can be said about the ontogenetic development in *E. barbara*. Excepting few descriptions based on tayra's development in captivity (*Poglayen-Neuwall, 1975*; *Poglayen-Neuwall & Poglayen-Neuwall, 1976*), no formal morphological description and quantification has been widely documented (but see *Schiaffini, 2020*).

This article analyzes the ontogenetic skull morphological changes throughout both sexes' postnatal development of *E. barbara* through qualitative and morphometric approaches. We also characterize the sex- and age-related functional differences in the masticatory biomechanical system. We discuss these differences and hypothesize their

possible ecological and evolutionary drivers. Little is known about intraspecific variation in tayras, and the same is true concerning its geographic variation. Traditionally, seven subspecies of *E. barbara* are recognized, but based almost exclusively on coat patterns (*Presley, 2000*; *Larivière & Jennings, 2009*). However, this division encounters little support regarding cranial characteristics (*Schiaffini, 2020*) and based on molecular evidence (*Ruiz-García et al., 2017*). Therefore, in this article, we focused our analyses to ontogenetic and sexual variation in this species as a whole.

## MATERIALS AND METHODS

### Data collection

Morphometric analyses were conducted with skulls of *Eira barbara* deposited in the collections of the American Museum of Natural History (AMNH; $n = 68$) in New York, the Field Museum of Natural History (FMNH; $n = 75$) in Chicago, the National Museum (USNM; $n = 35$) in Washington, DC, USA; Museum d'Histoire Naturelle (MNHN; $n = 16$) in Paris, France; Naturhistorisches Museum Wien (NMW; $n = 5$) in Vienna, Austria; Museu de Zoologia da Universidade de São Paulo (MZUSP; $n = 76$) in São Paulo, Museu Nacional (MN; $n = 32$) in Rio de Janeiro, Brazil; and Museu de Ciências Naturais da PUC Minas (MCN; $n = 7$) and Coleção de Mamíferos da Universidade Federal de Minas Gerais (UFMG; $n = 4$), both in Belo Horizonte, also in Brazil, totalizing 376 specimens (126 females, 133 males, and 117 individuals with undetermined sex), throughout *E. barbara* range and habitats in the Americas. The list of specimens studied is presented in Table S1.

We took 24 linear measurements with digital calipers (resolution 0.01 mm) (Fig. 1), of which 12 have functional meaning and were used as proxies in functional equations (see below). Measurement descriptions and acronyms are listed in Table 1.

Some of these measurements were used to estimate biomechanical parameters related to the masticatory performance. These functional equations are:

Temporal Muscle Size Estimation: $\boldsymbol{TMSE} = \left\{ ZW - \left[ \frac{BBC + POC}{2} \right] \right\} \times TFL$

Corrected Force-index Temporal muscle at canines: $\boldsymbol{CFTC} = \sqrt{\frac{\sqrt{TMSE} \times TMA}{JL} \times JHM_1}$

Corrected Force-index Masseter muscle at canines: $\boldsymbol{CFMC} = \sqrt{\frac{MFL \times MMA}{JL} \times JHM_1}$

The skulls were also surveyed for qualitative differences in tooth eruption, cranial suture closure patterns, and general consistent differences in skull appearance between specimens that could be related to differences among age categories and between the sexes. Since the actual stage of life was unknown for virtually all the material analyzed, age was determined by establishing Age-classes (*King, 1980*).

### Analyses

The quantitative analysis included univariate and multivariate approaches. Sexual dimorphism was tested through each Age-class by Student t-test or Mann-Whitney tests, according to frequency distribution criteria (normality, kurtosis, and skewness), homoscedasticity, and sample sizes. The significance level was 0.05.
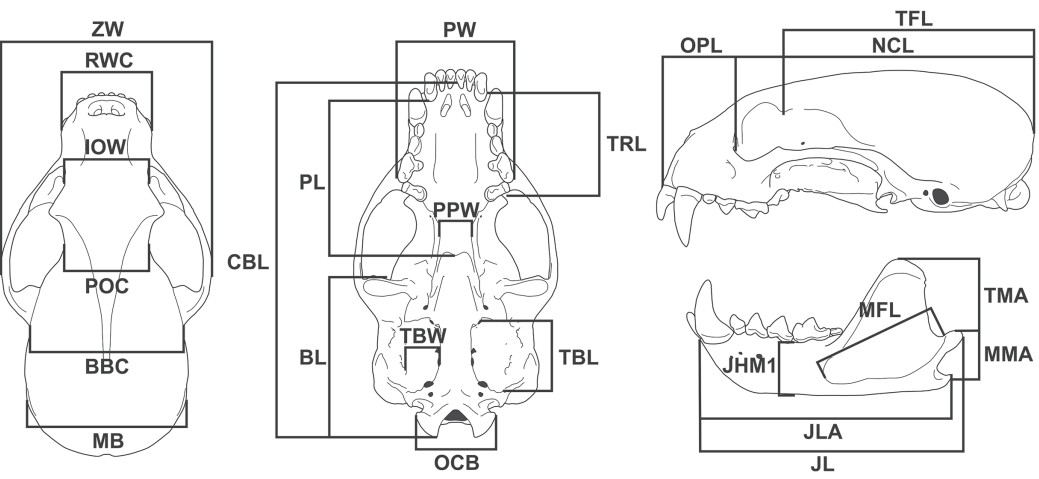

**Figure 1** Diagram showing the 24 linear measurements taken from the skull of *Eira barbara* used in this study.                                              

**Table 1 Acronyms and description of the linear skull measurements used in this study.**

|    | Skull measurement | Acronym | Description |
|----|-------------------|---------|-------------|
| 1  | Zygomatic width | ZW | Greatest width between the lateral surfaces of the zygomatic arches |
| 2  | Rostral width at the canines | RWC | Width between labial faces of upper canines |
| 3  | Interorbital width | IOW | Width at the anterior margin of the orbits |
| 4  | Postorbital constriction | POC | Width at the level of the postorbital constriction |
| 5  | Breadth of brain case | BBC | Width between the posterior base of the zygomatic process of squamosal |
| 6  | Mastoid breadth | MB | Width between the mastoid processes |
| 7  | Palatal width | PW | Width between the labial faces at the level of the metacone of the upper carnassials |
| 8  | Palatal length | PL | Length from the posterior margin of the alveolus of the third upper molar to the posterior nasal spine |
| 9  | Basicranial length | BL | Length from the anterior margin of the preglenoid process to the distal tip of occipital condyle |
| 10 | Upper toothrow length | TRL | Length from the anterior margin of the upper canine to the distal margin of the first upper molar at the level of the metacone |
| 11 | Post palatal width | PPW | Width between the entopterygoid processes |
| 12 | Tympanic bulla length | TBL | Length from the anterior carotid foramen to the posteriormost point of the entotympanic, at the level of the suture with the paracondylar processes |
| 13 | Tympanic bulla width | TBW | Width between the stylomastoid foramen and the posterior carotid foramen |
| 14 | Occipital condyles breadth | OCB | Width between the medial surface of the occipital condyles |
| 15 | Condylobasal length | CBL | Length from the anterior tip of the premaxilla to the distal tip of the occipital condyles |
| 16 | Orbit to premaxilla length | OPL | Length from the anterior margin of the orbit to the anterior tip of the premaxilla |
| 17 | Neurocranium length | NCL | Length from the anterior margin of the orbit to the most distal point of the supraoccipital |
| 18 | Temporal fossa length | TFL | Length from the tip of the postorbital process of the frontal to the most distal point of the supraoccipital |
| 19 | Temporal muscle moment arm | TMA | Length from the dorsal tip of the coronoid process to the posterior tip of the angular process |

(Continued)

| | Table 1 (continued) | | |
|---|---|---|---|
| | **Skull measurement** | **Acronym** | **Description** |
| 20 | Masseter muscle moment arm | MMA | Length from the dorsal tip of the condylar process to the posterior tip of the angular process |
| 21 | Jaw length | JL | Length from the anterior tip of the dentary to the posterior tip of the condylar process |
| 22 | Jaw length at angular process | JLA | Length from the anterior tip of the dentary to the posterior tip of the angular process |
| 23 | Jaw height at m1 position | JHM1 | Height of the dentary measured immediately anterior to first lower molar |
| 24 | Masseteric fossa length | MFL | Length between the anteriormost and posteriormost points of the masseteric fossa |

Principal component analysis (PCA) was used to characterize the overall morphological patterns of the sex- and age-groups. The objective was to describe the principal axes of variation and the distribution of males and females of the different age classes over the species' morphological space. Therefore, PCA allowed for the morphological trends that characterize tayra males' and females' ontogenetic development to be recognized in a multidimensional space.

Discriminant function analysis (DFA) was used to statistically differentiate sex- and age-groups. This method allows for a significance test of the overall difference between *a priori*-defined groups and the measurements that most contribute to distinguishing these groups. The squared Mahalanobis distance was used as a multivariate *post-hoc* method to characterize the differences between each group and the correspondent significance probability.

Morphological allometries between sexes over the development may incur functional differences with potentially different ecomorphological outcomes. The study of the skull functionality focused on the biomechanical systems of the two principal masticatory muscles: temporal muscle group and masseter muscle group. The pterygoid muscle system is assumed to have a function comparable to that of the masseteric system and was not included in the analysis. Morphometric proxies called force-indexes based on the static equilibrium equation ($\sum M = 0$, the sum of all force moments equals zero) were used to estimate the bite forces according to the individuals' sex and age (*Kiltie, 1982*; *Radinsky, 1987*; *Sicuro & Oliveira, 2002*, *2011*; *Sicuro, Neves & Oliveira, 2011*; *Sicuro et al., 2021*). Some studies comparing morphometrics of musculoskeletal features and *in vivo* direct measurements have demonstrated the validity of morphometric proxies to compare the functionality of jaw biomechanical systems of morphologically- or phylogenetically-related species (*Ginot et al., 2018*; *Sicuro et al., 2021*; *Herrel et al., 2024*). The force-indexes presented similar comparative power to the direct measurement of forces in *in vivo* experiments (*Sicuro et al., 2021*). The functionality of the masticatory apparatus has a marked adaptative meaning to predation and other behaviors, including territorial disputes and mating displays (*Turnbull, 1970*; *Kiltie, 1982*; *Radinsky, 1987*; *Greaves, 1995*; *Wainwright & Reilly, 1994*; *Hartstone-Rose, Hertzig & Dickinson, 2019*; *Sicuro & Oliveira, 2002*), being useful to conjecture ecomorphological interactions. Therefore, the Corrected

Force-index of Temporal muscle at canines is a functional morphological proxy related to the maximum bite force at the canines, where the in-force lever arm is the distance between the jaw articular process and the tip of the coronoid process, and the out-force lever arm is the distance of the same articular joint to the canine. The temporal muscle group area is given by the product between the Temporal fossa length and the breadth of the empty space between the parietal and the zygomatic arch. Finally, the jaw high at the $M_1$ is used as an indicator of the dentary bone robustness to improve the accuracy of the force-index by adding more morphological information. The same logic is applied to the Corrected Force-index Masseter muscle at canines, except the morphological information of the masseter muscle scar is used instead. The sex and age groups have their force-indexes compared through ANCOVA to assess the ontogeny of the temporal and masseteric biomechanical systems.

## RESULTS

### Qualitative analyses

Not all skull characteristics were useful to the establishment of age classes. For instance, the timing of closure for many sutures is highly variable in some individuals. Conversely, other characteristics, such as the sequence of teeth eruption, seem more consistently chronologically related. Based on these criteria, the specimens were classified into four age categories (Fig. 2; Table S1):

Class I, including 17 individuals (four females, 13 males): specimens in this age class have the postorbital constriction visibly wider than the interorbital constriction and no or little sign of parasagittal crests. All sutures, in particular, the frontal/parietal, nasal/maxillar/frontal, and basisphenoid/basioccipital sutures, are evident (Fig. 2), and only deciduous postcanine teeth are functional (although some individuals may already have erupting permanent teeth) (Fig. 3).

Class II, including 23 individuals (nine females, 14 males): specimens in this age class have the width of the postorbital constriction similar to that of the interorbital constriction, and parasagittal crests are, in most cases, clearly distinguishable. The frontal/parietal, nasal/maxillar/frontal, and basisphenoid/basioccipital sutures are still clearly visible (Fig. 2). Specimens have both deciduous and permanent teeth, in particular, the deciduous ($DP^3$) and permanent ($P^4$) carnassials are both functional simultaneously (Fig. 3).

Class III, including 37 individuals (18 females, 19 males): mostly specimens that could be considered subadults but with adult cranial dimensions, with postorbital constriction narrower than interorbital constriction, parasagittal crests well distinguishable, and frontal/parietal suture obliterated, although nasal/maxillar/frontal and basisphenoid/basioccipital sutures remain visible (Fig. 2). The dentition is fully permanent (Fig. 3).

Class IV, including 182 individuals (95 females, 87 males): Fully adult specimens, with full permanent dentition, postorbital constriction clearly narrower than interorbital constriction, all cranial sutures obliterated, and well-marked parasagittal crests, converging into sagittal crests in males (Fig. 2).
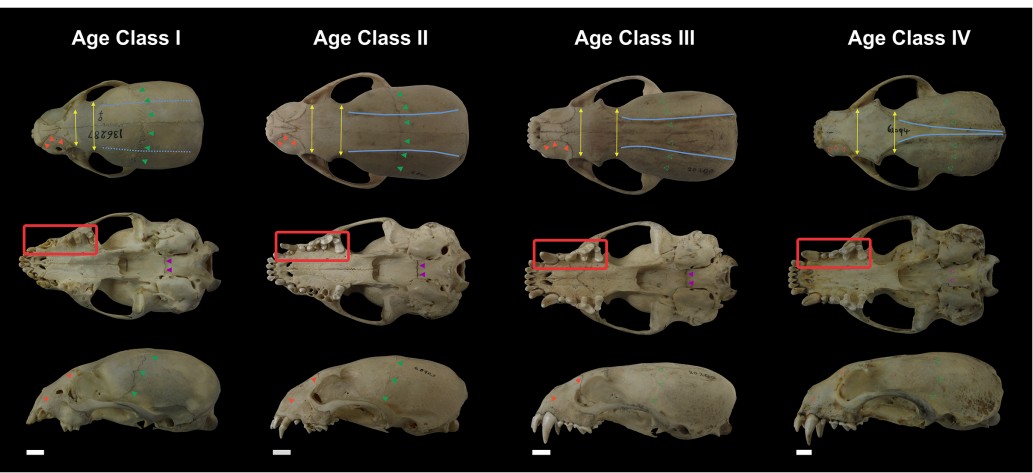

**Figure 2 Representative skulls of *Eira barbara* specimens classified in Age Class I (AMNH 136287), Age Class II (FMNH 68903), Age Class III (FMNH 20200) and Age Class IV (AMNH 62094).** The highlighted areas indicate the qualitative characters used to distinguish the different age classes, such as proportions between the postorbital and interorbital constriction (yellow), visibility of the frontal/parietal suture (green), visibility of the nasal/maxillar/frontal suture (orange), visibility of the basisphenoid/basioccipital suture (purple), presence of parasagittal and temporal crests (blue). Scale bars = 1 cm.

An actual age determination (*e.g.*, months, years) cannot be established for this age classification due to the lack of information associated with the specimens (see below in the Discussion).

Male and female skulls of tayra show, for the most part, little qualitative differences in earlier age classes and can only be positively separated once they reach Age Class IV (Fig. 4). At this age class, skulls of males, in general, tend to be larger and more robust than females, with wider rostra and zygomatic arches. The parasagittal crests tend to consolidate in sagittal crests in males, which rarely occur in females.

## Quantitative analyses

The evaluation of groups' data distributions found no major problems of kurtosis, skewness, and normality (Shapiro-Wilk test). The comparison between sexes in each age class indicated a progressive increase in sexual dimorphism over the ontogenetic development. Among the younger individuals (Age Class I), no significant differences were found in the 24 skull measurements (Man-Whitney test, $P > 0.05$). Sexual dimorphism starts in individuals of Age Class II, where 11 out of 24 skull measurements presented significant differences between male and female tayras (Student t-test, ranging from $P < 0.05$ to $P < 0.01$). Among Age Class III individuals, sexual dimorphism is already well marked, where 20 out of 24 skull measurements indicate significant differences between sexes (Student t-test, ranging from $P < 0.03$ to $P < 0.00001$). The older group (Age Class IV) presents a consolidated sexual dimorphism in all skull measurements assessed with a marked significance (Student t-test, $P < 0.00001$ in all variables). These analyses suggest that the sex-related differences grow allometrically according to the skull features, but the

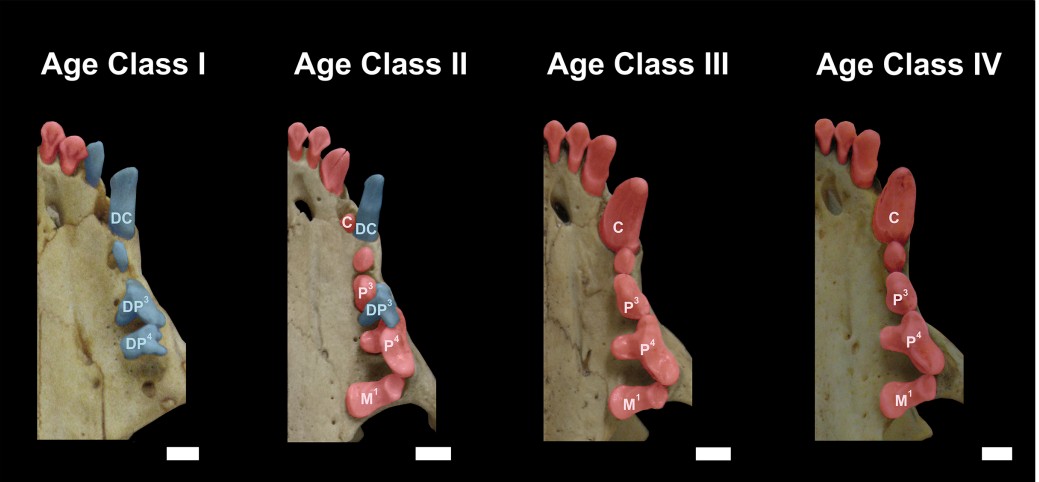

**Figure 3 Dentition of *Eira barbara* belonging to Age Class I (AMNH 98583), Age Class II (FMNH 68903), Age Class III (FMNH 20200) and Age Class IV (AMNH 62094).** Highlighted teeth denote the dental criteria used for distinguishing different age classes, as the prevalence of deciduous teeth (blue), in particular $DP^3$ and $DP^4$, in Age Class I, the co-occurrence of $DP^3$ and $P^4$ in Age Class II, and the occurrence only of permanent teeth (red) in Age Classes III and IV. DC = Deciduous Upper Canine; $DP^3$ = Deciduous Third Upper Premolar; $DP^4$ = Deciduous Fourth Upper Premolar; C = Permanent Upper Canine; $M^1$ = Upper First Molar; $P^3$ = Permanent Third Upper Premolar; $P^4$ = Permanent Fourth Upper Premolar. Scale bars = 0.5 cm.

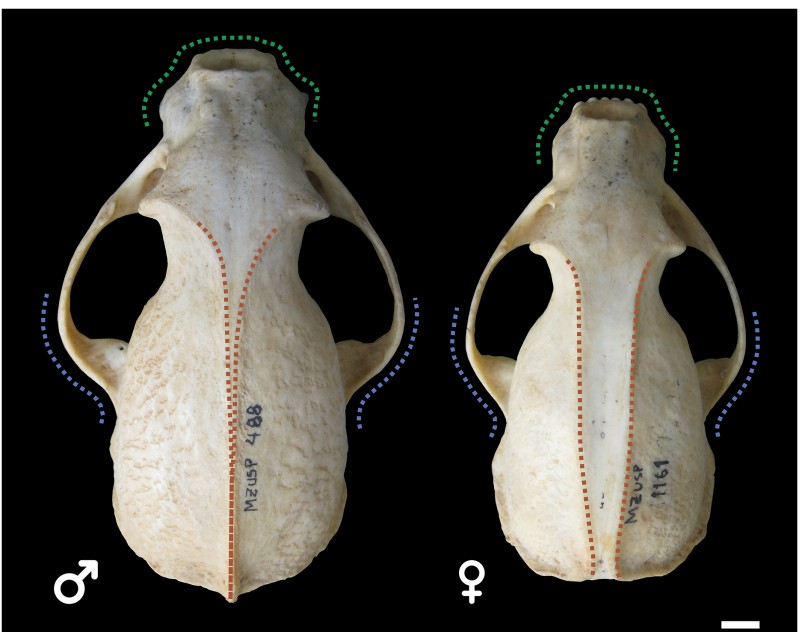

**Figure 4 Skulls of male (MZUSP 488, left) and female (MZUSP 1161, right) *Eira barbara*, both belonging to Age Class IV.** Diagrams show some qualitative distinguishing characteristics between males and females of *E. barbara*, such as the wider zygomatic arches in males, especially at the base (blue), wider rostrum in males (green), and the sagittal crest consolidation in males (orange), which rarely occur in females. Scale bar = 1 cm.

significance of the differences also grows deep. This fact means that the overlap between the statistical distribution of males' and females' skull features decreases, denoting an even more marked sexual dimorphism over the ontogeny. The complete statistical comparison between sexes over age classes is presented in Table S2.

PCA's first two principal components (PC1 and PC2) explain 78.2% of the total variation. Almost all skull measurements have high loads in the PC1, and the individual scores related to this PC follow a gradient from younger individuals to older ones. Therefore, the PC1 clearly expresses the size variation among the sex- and age-groups. On the other hand, in PC2, the contribution of the variables indicates two directions that denote opposite morphological trends. The PC2 positive side has a marked influence on the basicranial length (BL) and masseteric fossa length (MFL).

In contrast, the negative portion is influenced by the postorbital constriction (POC) and the post palatal width (PPW). Diversely from the PC1, which is more associated with the size of the individuals, the PC2 denotes aspects of the skull shape. The PC2 comprises information about the elongation of some skull features in the sagittal plane and the broadness of the central part of the skull (between the splanchnocranium and the neurocranium) in the coronal plane. Therefore, the groups whose scores distribute on the positive portion of the PC2 present more elongated basicranial regions and a large attach region of the masseter muscle group, while those groups on the negative portion of the PC2 present a broader postorbital constriction and broader post-palatal width, which are more pronounced from Age Class III onwards (Fig. 5). These overall trends corroborate the stages of morphological changes described in the qualitative approach.

Considering the full context of the morphological space defined by PC1 and PC2, younger individuals occupy the space defined by a small bulky skull with an enlarged postorbital portion. Along ontogenetic development, skulls of both male and female tayras become larger but with a marked narrowing of the postorbital constriction and palatal region. However, different sex-related allometric trends are observed where Age-class IV females consolidate a marked narrowing of this central part of the skull, while in Age-class IV males, this feature is less pronounced. Furthermore, Age-class IV *E. barbara* males keep growing and have bigger skulls than the same age-class females (Fig. 5).

Another aspect that could be inferred from the PCA is that not only the sexual dimorphism is already well marked from Age-class II onwards, but also that subadult females (Age-class III) occupy almost the same morphological space of the adult females (Age-class IV). The same is not observed among tayra males, which could be related to an eventual earlier sexual maturity of females.

Likewise, in the univariate analyses, the DFA confirmed the differences between sex- and age-groups. The overall differences were highly significant: Wilks' Lambda: 0.02 ~ $F_{168,1549} = 7.26$ $P < 0.0001$, and the squared Mahalanobis distances provided the specific *P*-values between groups (Table 2). The lack of significant sex-related differences between younger groups is also denoted in the DFA, as well as, the sexual dimorphism in skull features starts to be well marked between Age-class III and IV males and females.

The quantitative analyses of *E. barbara* skulls show that earlier in their development, there was no evident sexual dimorphism in tayras. Over ontogenetic development toward

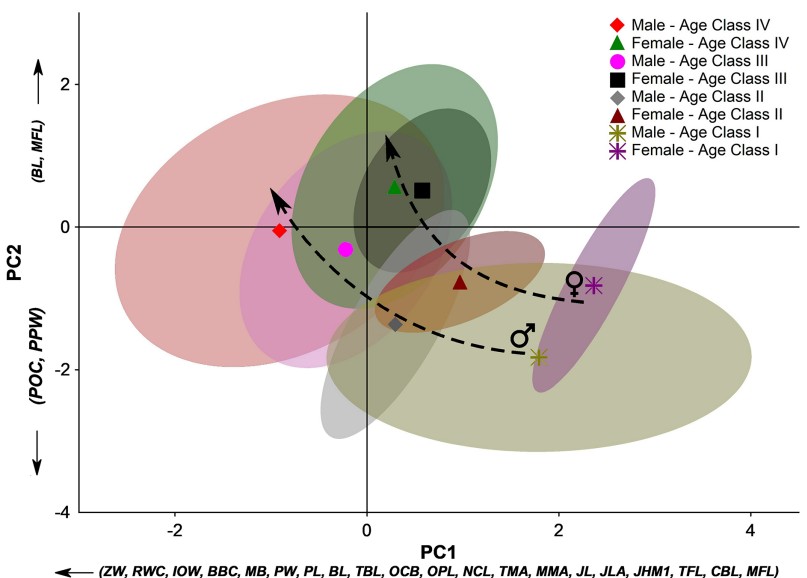

**Figure 5 Principal component analysis (PCA) of the 24 linear measurements taken from the skull of** *Eira barbara* **for each male and female age class.** The first two principal components (PC1 and PC2) explain 78.2% of the total variation. For linear skull measurement acronyms, see Table 1. The symbols denote the groups' centroids, and the colored areas represent 95% confidence ellipses for each dataset. Dashed arrows indicate different ontogenetic trends between sexes.

**Table 2 *P*-values of squared Mahalanobis distances between sexes and age groups of *Eira barbara*.** Significant values are marked with asterisks (***).

| | Male–Age class IV | Female–Age class IV | Male–Age class III | Female–Age class III | Male–Age class II | Female–Age class II | Male–Age class I | Female–Age class I |
|---|---|---|---|---|---|---|---|---|
| **Male–Age class IV** | – | 0.00000001*** | 0.00000001*** | 0.00000001*** | 0.00000001*** | 0.00000001*** | 0.00000001*** | 0.00000001*** |
| **Female–Age class IV** | 0.00000001*** | – | 0.00000001*** | 0.00001*** | 0.00000001*** | 0.00000001*** | 0.00000001*** | 0.00000001*** |
| **Male–Age class III** | 0.00000001*** | 0.00000001*** | – | 0.00001*** | 0.000001*** | 0.001*** | 0.00000001*** | 0.00000001*** |
| **Female–Age class III** | 0.00000001*** | 0.00001*** | 0.00001*** | – | 0.00000001*** | 0.000001*** | 0.00000001*** | 0.00000001*** |
| **Male–Age class II** | 0.00000001*** | 0.00000001*** | 0.000001*** | 0.00000001*** | – | 0.18 n.s. | 0.00000001*** | 0.00000001*** |
| **Female–Age class II** | 0.00000001*** | 0.00000001*** | 0.001*** | 0.000001*** | 0.18 n.s. | – | 0.00000001*** | 0.00000001*** |
| **Male–Age class I** | 0.00000001*** | 0.00000001*** | 0.00000001*** | 0.00000001*** | 0.00000001*** | 0.00000001*** | – | 0.32 n.s. |
| **Female–Age class I** | 0.00000001*** | 0.00000001*** | 0.00000001*** | 0.00000001*** | 0.00000001*** | 0.00000001*** | 0.32 n.s. | – |

sub-adult and adult stages, sexual morphological differences in the skull become more and more pronounced. These differences are size- and shape-related, mainly associated with the overall skull growth and the narrowing of the postorbital constriction and post-palatal

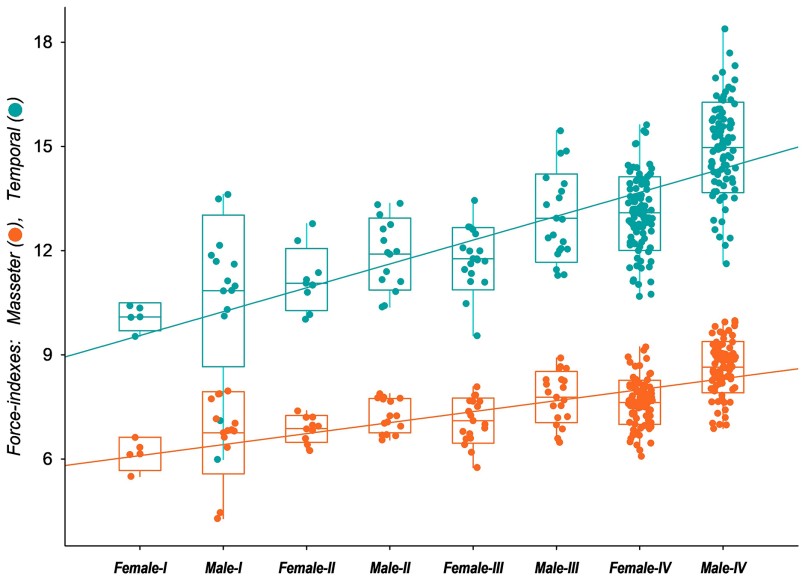

**Figure 6 Box-plot with jitter-function of the estimated values of bite force indexes (Corrected Force-index of Temporal muscle at canines, CFTC; Corrected Force-index of Masseter muscle at canines, CFMC) from the skull of *Eira barbara* for each male and female.**

width. In comparison, this narrowing trend of the postorbital constriction is even more striking in tayra females of Age-classes III and IV. In contrast, this feature in Age-class IV male individuals is comparatively broader than that of females.

The functional analysis highlights the possible influence of the distinct skull development paths among male and female tayras on the biomechanics of temporal and masseteric systems. The results of the force-indexes were plotted against the age classes representing the ontogenetic stages (Fig. 6). Since *E. barbara* shares the conspicuous Gulonine skull pattern, it follows *Turnbull*'s *(1970)* carnivore-shearing *bauplan*, with a primary contribution of the temporal muscle biomechanical system for the overall bite force. However, over the individuals' growth, the contribution of the temporal biomechanical system to bite force becomes more prominent–especially in Age-class IV males–than the masseteric biomechanical system. The ANCOVA applied to the two slopes confirms the significant difference between the intercepts ($P < 0.00001$, d.f. = 1.515). Moreover, the angular coefficient (or slope) of the temporal biomechanical system is also significantly higher than the masseteric ($P < 0.00001$, d.f. = 1.514). This result denotes the allometric development between the two muscle groups' biomechanical systems, where those related to the temporal muscle group become gradually more predominant to the bite force. In addition, there is an evident similarity of the temporal group force-index (CFTC) between Age-class IV females and Age-class III males, followed by a marked increase of the same CFTC in Age-class IV males. This bite force increment of tayra adult males is consistent with their superior size development.

## DISCUSSION

Many attempts have been made to determine age classes in mustelids in general and, particularly, in Guloninae (*e.g.*, *Martes martes*, *Martes foina*, *Habermehl & Röttcher, 1967*; *Reig & Ruprecht, 1989*; *Mustela putorius*, *Buchalczyk & Ruprecht, 1977*; *Mustela nivalis*, *Schmidt, 1992*). Fur-bearing species such as martens and fishers had their age classes especially well studied. However, this age determination is usually applicable only to specimens that have replaced all deciduous teeth and are regularly found outside the den (which may correspond to Age Classes III and IV in this article). In Holarctic martens, this usually represents individuals older than 8 months and caught in their first winter. We still know very little about the growth and when younglings reach independence in tayras. However, their Guloninae relatives (*i.e.*, those that live in more marked seasonal environments) show intense and well-determined bursts of growth, usually during the summer and spring, which account for the unequivocal characteristics that make it possible to determine with precision the differences among nestlings and juveniles which had left the den (*Heptner et al., 2001*). On the other hand, growth in tayras, once they reach Age Class III, seems to occur much less predictable, with no clear patterns until they reach complete obliteration of sutures in Age Class IV.

The challenge of determining the age of mustelids when known-aged material is unavailable is long-standing in wildlife management (*King, 1980*). Usually, some age classification based on growth and development series must be used. However, by its own nature, it is usually relative and does not help establish a definitive age unless these characteristics are directly associated with known age material. In the case of the tayra, almost none of the material studied here has a definitive age (*i.e.*, months or years) associated with it, and it represents a challenge to relate the Age Classes proposed here with actual growth stages and ages in tayras, given the lack of knowledge about the species' life history. *Poglayen-Neuwall & Poglayen-Neuwall (1976)* provide a list of weights and external measurements (besides extensive notes on behavior and vocalization development) of captive-born litters of tayras at the Louisville Zoological Garden (today Louisville Zoo). However, little information is provided on qualitative and quantitative traits that can be used to track physical changes during development in tayras.

Studies with temperate species of Guloninae show that they usually grow very fast, attaining adult size in only a few months (*Reig & Ruprecht, 1989*). Therefore, skulls older than 5 months usually are adult-sized. In wolverines (*Gulo gulo*), growth is almost complete between 9 and 12 months of age, but males keep growing and changing some cranial proportions after 1 year (*Wiig, 1989*). Martens of the genus *Martes*, including *M. martes* and *M. zibellina*, are usually thought to have attained adult size around four- to 6-months-old (*Brassard & Bernard, 1939*; *Marshall, 1951*), including the eruption of permanent teeth. However, many adult-sized marten individuals are still considered juveniles or sub-adults, and linear body measurements are usually considered prone to high misclassification rates for discrimination between emerging adults and sub-adults (*Ruette et al., 2015*). Wolverines also show fast postnatal growth, which may be a recurrent pattern in mustelids (*Wiig, 1989*).

The same may be true for tayras, and many adult-sized individuals classified here as Age Class III may not be sexually mature in fact. Tayras are thought to attain adult sizes by 6 months (*Encke, 1968*) and be sexually mature at 18 months (*Poglayen-Neuwall, 1975*; *Presley, 2000*). This condition is similar to what is known for species of the genus *Martes* and *Pekania*, which are usually adults after 18 months of age, and by 6 months, they have already replaced all of their deciduous dentition (*Bryant, McGillivray & Bartlett, 1997*; *Ruette et al., 2015*). They are usually born in early-middle spring, in March and April, considering the demands imposed by the well-marked seasons in temperate latitudes and the coming of winter. Little is known about tayra's reproductive cycle. However, it seems that, unlike Holarctic Guloninae, it is a non-seasonal breeder with a polyestrous cycle and the female in heat many times a year (*Poglayen-Neuwall et al., 1989*; *Larivière & Jennings, 2009*). Our data provide some support for this idea since individuals measured from Age Class I were collected almost over the whole year, including January (three specimens), March, April (two specimens), July (three specimens), August (four specimens), September, October, November (two specimens), and December. The fact that Age Class I individuals' skull measurements present low variation coefficients (Table S2) suggests that they have no major morphological divergences from the average when collected, supporting the idea that juveniles of tayra can be found throughout the year.

Qualitative characters represent a continuous growth series with considerable overlap between classes and difficulty in judging clear-cut characters. Erupting teeth are usually a reliable way of telling age in mammals. However, deciduous teeth are frequently replaced early during development (*Slaughter, Pine & Pine, 1974*) and are only helpful for very young animals. According to the tooth eruption table provided by *Poglayen-Neuwall & Poglayen-Neuwall (1976)*, based on five captive-born specimens of tayra, individuals of our first age category are probably less than 6-months-old. In other martens, such as sables (*M. zibellina*), the replacement of deciduous teeth by permanent ones begins at 3.5 months and may also be completed by 6 months (*Heptner et al., 2001*).

The closure of sutures, especially nasal sutures, is also sometimes used to establish age in mustelids (*Buchalczyk & Ruprecht, 1977*; *King, 1980*). However, sutures are also very variable in terms of timing of obliteration, and there is considerable overlap between previously established age classes (*Buchalczyk & Ruprecht, 1977*), so they may not be a reliable criterium to determine age in mustelids. Mustelids (especially martens) usually close their skull sutures very early (*Reig & Ruprecht, 1989*). In the tayra, the timing for closure of most sutures is highly variable in some individuals, frequently showing a mosaic of characteristics with other age indicators established here. Younger specimens are generally easily identifiable by showing all the sutures open, while older specimens have their skull sutures entirely obliterated. However, we found that the nasal/maxillar/frontal and, especially, the basisphenoid/basioccipital sutures obliterate relatively late, possibly related to the anteroposterior skull growth. The closure of the latter, in particular, was considered a sure way to establish the transition between Age Classes III and IV.

Another characteristic sometimes mentioned as a sign of sexual maturity in martens are sagittal crests, which seem to have a marked consolidation pattern throughout the seasons, culminating with its formation from the parasagittal crests in later age categories

(*Habermehl & Röttcher, 1967*; *Reig & Ruprecht, 1989*; *Heptner et al., 2001*). In tayras, sagittal crests are more commonly formed in males than females, which agrees with what is known for martens (*Habermehl & Röttcher, 1967*). However, in contrast with true martens, sagittal crests do not seem to have a clear formation pattern in tayras. They are much more variable and have no evident relationship with other ontogenetic markers analyzed here, although they are vastly more prevalent in Age Class IV. The absence of a regular and seasonal collection of tayras and the lack of information about the growth and development of this species preclude further refinement of the age categories. The utmost evidence of the validity of the age classification proposed here and the individuals' actual age could only be achieved with a direct chronological connection between *E. barbara* living individuals and their collected skulls. For now, we consider that the presence of the sagittal crest is not a valuable feature to establish age in *E. barbara*, being present and consolidated mostly in some older male specimens.

The increase in zygomatic breadth and narrowing of the postorbital constriction are also other frequently used morphological characters to allocate species of Guloninae and other mustelids to age classes (*Wright & Coulter, 1967*; *Heptner et al., 2001*; *Özen, 2020*). However, classes defined based on this feature usually overlap considerably (*King, 1980*). In tayras, the narrowing of the postorbital constriction is noticeable in Age Class II, and by Age Class III, it is well marked.

Some of these characters are sexually dimorphic in *E. barbara* during development. Sexual dimorphism in skull proportions is widespread in both martens and other mustelids (*Buchalczyk & Ruprecht, 1977*; *Wiig, 1989*; *Schmidt, 1992*), although not to such extremes as found in some other carnivores (*Gittleman & Valkenburgh, 1997*; *Lindenfors, Gittleman & Jones, 2007*). Even though it is not as well marked as in other species of Mustelidae (*Presley, 2000*), the sexual dimorphism in tayras is still present and occurs along similar lines to that of previously studied Holarctic species. *Noonan et al. (2016)* found that mustelids with a more carnivore diet tend to have higher sexual dimorphism than more generalist ones, which may be related to resource partitioning between individuals with highly patched resource sources. The generalist habits of the tayra, together with its more predictable tropical habitat, may account for the relatively low sexual dimorphism compared to some other Guloninae. Differences between males and females in tayras are mainly quantitative, related to the greater size and robustness of the males' skull, with greater sagittal crest development, which may indicate hypertrophied temporal muscles.

Sexual differences in Guloninae skulls are usually expressed as vague changes in shape, increase in general size, and elongation of the skull (*e.g.*, *Heptner et al., 2001*), testifying for the relatively low sexual dimorphism in the whole group, when compared to other carnivorans. A possible exception may be *G. gulo*, which is usually thought to show remarkable sexual differences, with males being substantially larger and more robust than females (*Wiig, 1989*; *Heptner et al., 2001*), and fishers (*P. pennanti*), which may show the strongest sexual dimorphism among Guloninae, with measurements of males being significantly larger than females, even in juveniles (*Wright & Coulter, 1967*; *Bryant, McGillivray & Bartlett, 1997*), and with some males in special showing remarkably

pronounced sagittal crests. Even so, superposition between the sexes is widespread within the group. *Law & Mehta (2018)* found that, among the Guloninae, only *P. pennanti* presents significant sexual dimorphism in both skull size and cranial shape. Notwithstanding, bite force differences are common. For instance, males of *P. pennanti*, *G. gulo*, and *Martes flavigula* present stronger bites than females. In comparison, *E. barbara* is in the lower range of sexual dimorphism within the group, having relatively modest levels of difference in size and shape despite of the marked statistical sex-related difference in adults' bite force.

Our functional analysis highlights the possible influence of the distinct skull development paths among male and female tayras on the biomechanics of temporal and masseteric systems. Over the individuals' growth, the contribution of the temporal biomechanical system to bite force becomes more prominent. Tayras show little sexual dimorphism in the early stages of life, but males and females develop distinct characteristics in later stages, mainly due to heterochrony in the development of males. It is often assumed that, in other Guloninae, the differences between adult males and females can be attributed to different growth trajectories (*Heptner et al., 2001*), with sutures closing later in males–due to continuous growth–than in females. In *Martes martes* and *M. foina*, it also seems that males keep growing diversely from females, which would explain the retention of more juvenile proportions in adult females (*Reig & Ruprecht, 1989*). It has been theorized that the differences in skull shape between males and females in mustelids are due to the retention of juvenile characters in females (*King, 1980*). In tayras, we hypothesize that males with stronger temporal musculature and more powerful bite forces may, at least in part, be evolutionarily selected due to demands of territorial defense or mating behavior, since males grab and hold females by the neck to subdue them during copulation (*Poglayen-Neuwall, 1975*; *Fuentes Magallón et al., 2021*).

The mainstream assumption is that most carnivores show some degree of divergence in body size and cranial morphology between sexes–usually biased towards males' musculoskeletal hypertrophy–and that these differences are attributable to sexual selection (*Isaac, 2005*; *Law & Mehta, 2018*). As in most carnivores, martens (including *E. barbara*) show sexual dimorphism in which males have larger and heavier skulls (*Allen, 1938*). A frequently evoked hypothesis to explain sexual dimorphism in carnivores is that females and males may be under different selective pressures related to, for instance, polygynous mating systems (*Moors, 1980*; *Holmes & Powell, 1994*). In this context, larger and stronger males are favored in securing the highest reproductive success due to their ability to defend wider-ranging territories, which include many reproductive females.

Most Guloninae species are indeed solitary and show a polygynous mating system, with a single male defending territories that include up to two or three females (*Wiig, 1989*; *Herrmann, 1994*; *Powell, 1994*; *Schröpfer, Wiegand & Hogrefe, 1997*; *Xu et al., 1997*; *Weir, Harestad & Corbould, 2009*). The elusive yellow-throated marten seems to be an exception, sometimes even considered monogamous (*Proulx & Aubry, 2017*). On the other hand, despite the scarce information about the tayra, it is suggested that it is promiscuous and has many yearly estruses (*Proulx & Aubry, 2017*). Male tayras do not remain with females

after parturition and have relatively large and overlapping territories. Due to this behavior, some may consider it less territorial than other species of mustelids (*Presley, 2000*). Data is still scarce, but home ranges seem to vary significantly according to surveys based on a few tracked individuals (*Larivière & Jennings, 2009*; *Marques, Fernández-Montejo & Villafañe-Trujillo, 2021*). In this aspect, there is a clear contrast between the tayra and most other species of Guloninae occurring in temperate environments. It is noteworthy that the only other tropical Guloninae, *Martes flavigula*, also seems to show large territories and significant overlap between them (*Grassman, Tewes & Silvy, 2005*), which suggests that these deviations from the polygynous system found in most martens may be related to resource availability in tropical environments. However, too little is known about the natural histories of both *M. flavigula* and *E. barbara* to draw more significant conclusions about the role of sociability in sexual dimorphism in these species and an eventual connection to tropical environments.

Moreover, sexual dimorphism in Guloninae has been attributed to intraspecific competition between sexes, which may explain why its magnitude varies between different species in some studies (*Dayan et al., 1989*; *Ruette et al., 2015*). Morphological structures, especially those related to feeding, such as tooth and jaw characteristics, may potentially weigh more toward a sexual morphological differentiation when resource partitioning between sexes is a crucial driver (*Holmes & Powell, 1994*). However, life history data is rarely available to support these hypotheses, which is the case with *E. barbara*. Evidence for niche partitioning between sexes in martens is limited (*Holmes & Powell, 1994*), as is for mustelids in general, being probable that this is not a significant factor in determining sexual dimorphism (*Fairbairn, 1997*), and the same is also likely true for *E. barbara*.

There is also the possibility that the presence of sexual dimorphism is associated with phylogenetic history. Sexual dimorphism, in different degrees, is evident in mustelids (*Holmes & Powell, 1994*) but is also widespread in other Musteloid carnivores (*Macdonald & Newman, 2017*), suggesting that pronounced sexual size dimorphism is probably plesiomorphic for extant Mustelidae and occurs in these taxa as a result of inheritance from the common ancestor of the clade (*Bryant, McGillivray & Bartlett, 1997*). Therefore, phylogenetic inertia may be an important factor responsible for sexual dimorphism maintenance in mustelids in general (*Cheverud, Dow & Leutenegger, 1985*).

## CONCLUSIONS

In general, ontogenetic and sexual dimorphism patterns in tayras agree with what is known for other Guloninae. Discrepancies may be attributed to differences in the life history of the tayra in tropical environments, but there is still little information about its ecology to establish clear correlations. As stated above, despite skull variation studies being ubiquitous for northern hemisphere species, particularly fur-bearing species, similar studies with tropical species–particularly Neotropical species–are uncommon. In this sense, this work presents the soundest overview of tayras' skull morphological and functional ontogenetic development hitherto. The data presented here

will contribute to further research in *E. barbara* geographic variation, ecology, and life-history studies.

## ACKNOWLEDGEMENTS

We would like to extend our deep appreciation for the collection curators and managers who allowed the examination of the specimens under their care: M.A. Eleanor Hoeger (AMNH), Dr. Bruce Patterson (FMNH), M.A. Darrin Lunde and M.A. Teresa Hsu (USNM), Dr. Géraldine Veron (MNHN), Dr. Alexander Bibl (NMW), Dr. Joyce Prado (MZUSP), Dr. João Alves (MN), and Dr. Claudia Costa (MCN). FLS also thank Dr. Norma Albarello and Dr. Oscar Rocha-Barbosa (IBRAG/UERJ) and Dr. Eduardo Ferreira (CESAM/UA) for all their support.

### Funding

Fernando Lencastre Sicuro received financial support from UID Centro de Estudos do Ambiente e do Mar (CESAM) +LA/P/0094/2020 through the Portuguese Foundation for Science and Technology, FCT/MCTES. The funders had no role in study design, data collection and analysis, decision to publish, or preparation of the manuscript.

### Grant Disclosures

The following grant information was disclosed by the authors:
FUID Centro de Estudos do Ambiente e do Mar (CESAM) +LA/P/0094/2020.
Portuguese Foundation for Science and Technology, FCT/MCTES.

### Competing Interests

The authors declare that they have no competing interests.

### Author Contributions

- Fernando Araujo Perini conceived and designed the experiments, performed the experiments, analyzed the data, prepared figures and/or tables, authored or reviewed drafts of the article, and approved the final draft.
- Fernando L. Sicuro conceived and designed the experiments, performed the experiments, analyzed the data, prepared figures and/or tables, authored or reviewed drafts of the article, and approved the final draft.

### Data Availability

   The raw data is available in the Supplemental Files.

### Supplemental Information

Supplemental information for this article can be found online at http://dx.doi.org/10.7717/peerj.19730#supplemental-information.

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
