# Peer review of "Sex- and age-related morphological and functional differences in the skull of Eira barbara (Linnaeus, 1758) (Carnivora, Mustelidae)"

_PeerJ, doi:10.7717/peerj.19730_

## Round 0.1 · original submission · Major Revisions

Dear Dr. Perini, I ask you to carefully clarify and supplement the manuscript in accordance with the comments of the reviewers. I hope that your responses to the reviewers will allow the manuscript to be approved for publication.

Reviewer 1 ·

Basic reporting

-

Experimental design

-

Validity of the findings

-

Additional comments

This manuscript provides important information on the cranial ecomorphology of the tayra, Eira barbara, adding relevant data on the morphofunctional and sexual variations that occur during the species' development. It represents a meaningful contribution to the understanding of morphological adaptations in a species that remains relatively understudied when compared to other mustelids. I have some comments regarding both minor and major points that I believe should be addressed prior to the publication of this manuscript.

Major points

The introduction is well written and structured, presenting a brief description of the importance of investigations into development and sexual dimorphism in carnivores, particularly mustelids. However, some points should be highlighted and addressed before the manuscript is published.

- The authors state that "...sex differences incur in marked divergent aspects of their biology." (Lines 52–53) and "Some species have marked differences between sexes, while others seem to have no differences at all" (Lines 54–55), but they do not provide examples of cases where these conditions occur. Thus, these sentences remain subjective and do not facilitate the reader’s understanding. I suggest that the authors include and explain at least one example for each statement.

- The authors discuss the economic importance of mustelids (Lines 63–76), exemplifying the use of their coat by the fur industry and commenting on the contribution of accurate age and sex determination during harvesting and collection periods. I believe this paragraph would benefit from a brief discussion on the value attributed to mustelids with distinct fur coloration, emphasizing the artificial production of color variants in captivity (e.g., "sapphire mink") and how the commercial value of rare colorations can directly influence the abundance of wild populations, such as the increased hunting pressure on white-coated stoats during winter. This article may assist the authors in developing this discussion: https://www.mdpi.com/2076-2615/14/23/3354

- Still regarding mustelid coloration, traits such as the coat color patterns of tayras have been used as markers of intraspecific variation, with seven subspecies currently recognized based on their coloration and body size (Presley 2000). The validity of these subspecies is sometimes considered questionable (e.g., Schiaffini 2020), and this discussion should be at least briefly mentioned by the authors. It is also important to highlight that the authors indicate greater skull robustness and cranial musculature in males compared to females during development, but they do not clarify whether these variations occur differently among specimens from the distinct ecoregions where these subspecies are found.

- The authors comment (Lines 93–95): "Consequently, compared to other Guloninae, many aspects of its ecology and life history are still relatively unknown, except for a few isolated descriptions of captive tayras’ external physical and behavioral ontogenetic development." This sentence, in its current form, may be misleading. There are relevant investigations into the ecomorphology of tayras (e.g., https://link.springer.com/article/10.1016/j.mambio.2016.06.002), for example, which extend beyond descriptions of captive individuals. It appears that the authors intended to refer specifically to the lack of investigations on the life history of tayras. Therefore, I suggest removing the term "ecology" to avoid potential misinterpretation.
-
Minor points

- Line 59: Use of "and" instead of "&" in Sicuro and Oliveira, 2015

- Line 82: Linnaeus 1758 without a comma separating the author and year, and a double parenthesis after the year

- Ruette et al. 2005 is cited in the manuscript (Line 61), but in the references it is listed as Ruette et al. 2015

- Heptner 2001 is cited in references but is also cited with the years 2022 and 2002 in several parts of the manuscript. Check.
Heptner 2022 - line 459
Heptner 2002 - Line 90, 99, 356, 406, 424, 438, 456, 476
Heptner 2001 – Reference

- "Sicuro, Neves & Oliveira, 2011" is cited in the text (Line 168) and then cited as "Sicuro, Neves & Oliveira, 2021" (Line 171), which is correct?

- Figure 2: I suggest increasing the image quality.

- Figure 3: I suggest reducing the width of the scale bar.

- Figure 4: The images are shifted to the left. Please correct.

After these corrections, I recommend the article for publication.

Reviewer 2 ·

Basic reporting

Interesting research, a couple of references are needed (I suggested some).

I understand that some analyses are not possible due to a lack of information when the specimens were collected in situ (as indicated at L216).

Experimental design

The references used to estimate the bite forces according to the individual sex and age were obtained from studies with Tayassuidae and Felidae.

Please explain why those references and parameters are suitable for mustelids.

Validity of the findings

-

Additional comments

All my comments and suggestions are marked within the PDF file.

Annotated reviews are not available for download in order to protect the identity of reviewers who chose to remain anonymous.

Reviewer 3 ·

Basic reporting

No comment.

Experimental design

No comment.

Validity of the findings

No comment.

Additional comments

The study is well-writen and performed. The authors well exposed the knowledge gap investigated in Eira barbara and their results, obtained through a varied and robust methodology, filled the highlighted gap. The references used by the authors are appropriate for the topic of the study. The methods, as mentioned earlier, are well described and appropriate for the study. The authors used robust statistical tests to test their hypotheses and achieve the objectives of their research.
However, in lines 105-107 the authors mention characterizing the sex- and age-related functional differences in the masticator biomechanical system. It would be interesting to address something of this topic previously in the introduction to contextualize the objective of this approach in the study.

---

## Round 0.2 · Minor Revisions

Dear Dr. Perini, Please make some minor corrections and your article will be accepted for publication.

Reviewer 1 ·

Basic reporting

No comment

Experimental design

No comment

Validity of the findings

No comment

Additional comments

No comment

Reviewer 2 ·

Basic reporting

'no comment'

Experimental design

'no comment'

Validity of the findings

'no comment'

Additional comments

The manuscript still needs few corrections, All my observations and suggestions are highlighted within the pdf file.

Annotated reviews are not available for download in order to protect the identity of reviewers who chose to remain anonymous.

---

## Round 0.3 · accepted · Accept

Dear Dr. Perini, I congratulate you on the acceptance of this article for publication and hope for continued cooperation with our journal. I wish you further success in studying this topic.

Reviewer 2 ·

Basic reporting

no comment

Experimental design

no comment

Validity of the findings

no comment

Additional comments

no comment